# Short-term modulation of the lesioned language network

**Gesa Hartwigsen[1]\*, Anika Stockert[2], Louise Charpentier[1], Max Wawrzyniak[2], Julian Klingbeil[2], Katrin Wrede[2], Hellmuth Obrig[3], Dorothee Saur[2]**

[1]Lise Meitner Research Group Cognition and Plasticity, Max Planck Institute for Human Cognitive and Brain Sciences Leipzig, Leipzig, Germany; [2]Language and Aphasia Laboratory, Department of Neurology, University of Leipzig Medical Centre, Leipzig, Germany; [3]Clinic for Cognitive Neurology, University of Leipzig Medical Centre & Max Planck Institute for Human Cognitive and Brain Sciences, Leipzig, Germany

**Abstract** Language is sustained by large-scale networks in the human brain. Stroke often severely affects function and network dynamics. However, the adaptive potential of the brain to compensate for lesions is poorly understood. A key question is whether upregulation of the right hemisphere is adaptive for language recovery. Targeting the potential for short-term reorganization in the lesioned brain, we applied 'virtual lesions' over left anterior or posterior inferior frontal gyrus (IFG) in post-stroke patients with left temporo-parietal lesions prior to functional neuroimaging. Perturbation of the posterior IFG selectively delayed phonological decisions and decreased phonological activity. The individual response delay was correlated with the upregulation of the lesion homologue, likely reflecting compensation. Moreover, stronger individual tract integrity of the right superior longitudinal fascicle was associated with lesser disruption. Our results provide evidence for functional and structural underpinnings of plasticity in the lesioned language network, and a compensatory role of the right hemisphere.

**\*For correspondence:**
hartwigsen@cbs.mpg.de

**Competing interests:** The authors declare that no competing interests exist.

## Introduction

Language is sustained by large-scale, left-lateralized neural networks in the human brain. Stroke-induced lesions to key regions for language often severely affect network dynamics and cause language impairment. Accordingly, recent work considers aphasia a network disorder, emphasizing the importance of disturbed interactions between key areas (*Corbetta et al., 2015*; *Fridriksson et al., 2016*). However, the way the language network compensates for brain lesions is poorly understood. A controversial debate pertains to the recruitment of contralesional right hemisphere regions (*Chrysikou et al., 2013*; *Hamilton et al., 2011*; *Hartwigsen and Saur, 2019*; *Turkeltaub, 2015*). Some studies found that (early) upregulation of the right hemisphere was beneficial for aphasia recovery (e.g. *Crinion and Price, 2005*; *Robson et al., 2014*; *Saur et al., 2006*; *Weiller et al., 1995*). In contrast, other results showed that such upregulation reflected maladaptive changes in patients with non-fluent aphasia (e.g. *Heiss et al., 2013*; *Martin et al., 2005*; *Naeser et al., 2005*; *Naeser et al., 2011*; *Postman-Caucheteux et al., 2010*; *Rosen et al., 2000*). Accordingly, suppression of right-hemispheric 'overactivation' with inhibitory transcranial magnetic stimulation in patients with post-stroke aphasia after left-hemispheric lesions resulted in improved language production (e.g. *Barwood et al., 2013*; *Naeser et al., 2005*; *Naeser et al., 2011*; *Weiduschat et al., 2011*). Indeed, language reorganization appears to be a dynamic process and the contribution of the right hemisphere may change across the time course of recovery (*Hartwigsen and Saur, 2019*; *Saur et al., 2006*).

We were interested in elucidating rapid short-term plasticity in response to 'virtual lesions' in the lesioned language network to investigate the compensatory capacity of language regions. Virtual lesions provide a well-controlled approach to study short-term reorganization at the neural network level unconfounded by long-term recovery (*Siebner and Rothwell, 2003*). The few existing studies on short-term plasticity in the language domain were almost exclusively performed in healthy volunteers. Consequently, it remains unclear how the lesioned network reacts to focal perturbation of a key language node. The most parsimonious hypothesis is that virtual lesions increase the overall 'lesion load' in the lesioned brain. This should result in a shift of task-related activity to the right hemisphere, if compensatory upregulation of homologous regions supports language function (*Hartwigsen, 2018*). Beyond our interest in identifying basic mechanisms of short-term plasticity, the identification of such mechanisms is a prerequisite for devising novel therapeutic strategies, including the application of non-invasive brain stimulation.

We focused on two essential processes for successful every-day communication: phonological and semantic analysis. Previous functional neuroimaging studies in healthy volunteers have demonstrated that both processes engage differential parts of the left-hemispheric language network (see *Vigneau et al., 2006* for a meta-analysis). Specifically, aside from temporal regions, phonological processing relies on the posterior inferior frontal gyrus (pIFG) and supramarginal gyrus (SMG) (*Devlin et al., 2003*; *Poldrack et al., 1999*; *Price et al., 1997*). In contrast, semantic processing consistently engages anterior inferior frontal gyrus (aIFG), anterior temporal lobe (ATL), posterior middle temporal gyrus (pMTG), and angular gyrus (AG) (*Binder et al., 2009*; *Devlin et al., 2003*; *Vigneau et al., 2006*). In line with these results, studies on patients with structural lesions provided evidence for a key contribution of distinct parietal and frontal regions to phonological and semantic processing (*Corbett et al., 2009*; *Dewarrat et al., 2009*). In a similar vein, neurostimulation studies using inhibitory transcranial magnetic stimulation (rTMS), demonstrated the functional relevance of left pIFG and SMG for phonological processing (*Gough et al., 2005*; *Hartwigsen et al., 2010*; *Oberhuber et al., 2016*; *Romero et al., 2006*). Other rTMS studies further showed that the left aIFG, ATL, pMTG and AG contribute to semantic processing (*Binney and Ralph, 2015*; *Davey et al., 2015*; *Gough et al., 2005*; *Jung and Lambon Ralph, 2016*; *Whitney et al., 2012*). Based on these findings, we recently combined different inhibitory rTMS protocols over selected inferior frontal and parietal regions to demonstrate that phonological and semantic processing are integrated in two segregated functional-anatomic networks that connect inferior parietal and inferior frontal regions (*Hartwigsen et al., 2016*). Moreover, TMS-induced disruption of left AG or SMG in

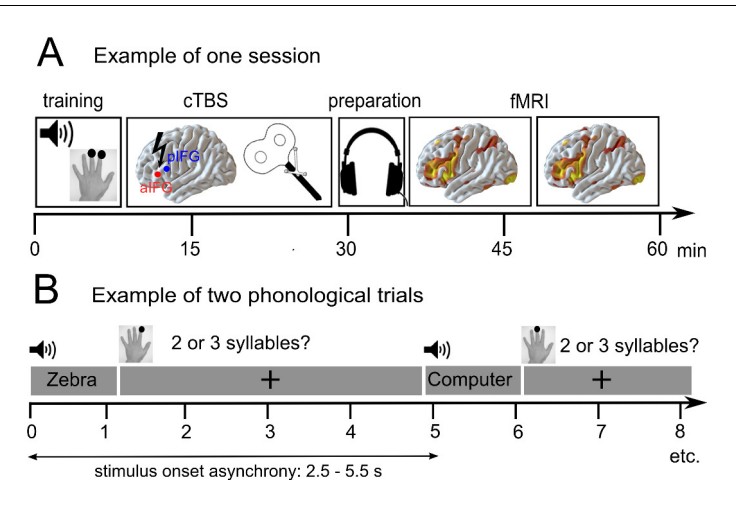

**Figure 1.** Overview of the experimental design. (**A**) After a short training session, patients received effective or sham continuous theta burst stimulation (cTBS-600) either over the anterior or posterior inferior frontal gyrus (a/pIFG) in different sessions. Thereafter, they performed phonological and semantic word judgement tasks in two fMRI runs. (**B**) Tasks were divided into 10 miniblocks per task and run, each consisting of 6 German words (e.g. 'Zebra' or 'Computer') with varying stimulus onset asynchrony. min = minutes; s = seconds.

**Table 1.** Patient characteristics: Test data.

| Patient | Age | Sex | Education | Laterality | Time since stroke | Lesion site | Lesion size (cm³) | % overlap with AG | % overlap with SMG | AAT |
|---|---|---|---|---|---|---|---|---|---|---|
| 01 | 54.2 | m | 12 | 0.9 | 10.1 | temporo-parietal | 66.90 | 76 | 40 | Isolated paraphasias comprehension deficits |
| 02 | 52.9 | m | 12 | 0.8 | 17.7 | parietal | 59.28 | 19 | 74 | No aphasia, expressive phon. deficits |
| 03 | 63.9 | m | 8 | 1 | 76.1 | parietal | 93.19 | 94 | 60 | Residual aphasia, expressive phon. deficits |
| 04 | 49.6 | m | 10 | 0.9 | 122.4 | parietal | 76.47 | 85 | 79 | Residual aphasia, expressive phon. deficits |
| 05 | 43.2 | w | 10 | 1 | 57.8 | parietal | 70.18 | 33 | 82 | Residual aphasia |
| 06 | 65.2 | m | 12 | 1 | 36.2 | parietal | 18.55 | 2 | 28 | No aphasia |
| 07 | 72.8 | w | 10 | 1 | 24.0 | parietal | 176.34 | 86 | 82 | Residual aphasia, decreased verbal memory span |
| 08 | 66.2 | w | 8 | 1 | 6.2 | temporo-parietal | 37.99 | 0 | 55 | Residual aphasia |
| 09 | 61.9 | m | 8 | 1 | 6.6 | temporo-parietal | 11.89 | 3 | 2 | No aphasia, decreased verbal memory span |
| 10 | 46.0 | m | 10 | 1 | 23.8 | parietal | 105.42 | 97 | 36 | Residual aphasia |
| 11 | 63.8 | w | 12 | 1 | 14.5 | parietal | 37.97 | 20 | 12 | No aphasia, decreased verbal memory span |
| 12 | 63.8 | m | 10 | 0.9 | 24.6 | temporo-parietal | 25.66 | 2 | 36 | No aphasia, decreased verbal memory span |

Laterality = Handedness (Oldfield score), % overlap = lesion overlap with angular gyrus (AG; BA 39) or supramarginal gyrus (SMG, BA 40); Education is given in years, Time since stroke in months. AAT = Aachener Aphasie Test (Aaachen Aphasia Inventory). phon = phonological.

the healthy brain resulted in large-scale inhibitory effects and increased the functional interaction between these networks (*Hartwigsen et al., 2017*). However, it is less clear how these networks interact in the lesioned language network.

To investigate the compensatory potential of these networks in the lesioned brain, we selected a sample of relatively well-recovered chronic stroke patients with lesions in the left temporo-parietal cortex and intact frontal areas. In our sample, 7 patients had residual aphasia and the remaining 5 showed complete recovery at the time of the experiment (see *subsection Patients* below and *Table 1* for details). Our rationale was that a virtual lesion induced by inhibitory continuous theta burst stimulation (cTBS; *Huang et al., 2005*) should uncover otherwise compensated deficits. This enabled us to differentiate between compensatory long-term reorganization and short-term effects of our intervention. The inclusion of two tasks (semantic and phonological word decisions) and two cTBS sites (aIFG and pIFG) allowed us to probe the potential for short-term reorganization in two major networks for language comprehension. The inclusion of a sham cTBS condition allowed for a direct within-subject comparison of effective and placebo cTBS, avoiding differences in task difficulty between patients and healthy controls.

We hypothesized that disruption of a task-specific subregion in the left pIFG should impair phonological performance, and disruption of aIFG should impair semantic performance; and both should inhibit task-related activation in the respective area. In turn, we investigated whether we might observe compensatory upregulation of (homologous) right-hemispheric regions or neighboring left-hemispheric regions that would support task processing after an 'acute' virtual lesion (*Binney and Ralph, 2015*; *Hallam et al., 2016*; *Hartwigsen et al., 2017*; *Hartwigsen et al., 2013*; *Jung and Lambon Ralph, 2016*). The inclusion of diffusion MRI data allowed us to investigate whether compensatory upregulation of homologous regions would be mediated by the underlying white-matter

connectivity in long-range fiber tracts of the intact right hemisphere. The combination of cTBS and neuroimaging data should provide a comprehensive characterization of task-specific short-term reorganization in the language network in patients with temporo-parietal lesions. An illustration of the experimental design is provided in *Figure 1*.

## Results

### Differential effects of cTBS on phonological and semantic response speed

Lesion overlap and cTBS sites are illustrated in *Figure 2A and B* (see also *Table 1* for patient characteristics). We found a functional-anatomical double dissociation of task and cTBS site for the mean response times (*Figure 2C*, *Table 2* for the mean data). Accordingly, the interaction between task and cTBS was significant ($F_{2,22} = 8.59$, p=0.002). Bonferroni-Holm corrected post-hoc paired t-tests showed that cTBS over pIFG significantly delayed phonological response times when compared against cTBS over aIFG ($t_{11} = 3.19$, p=0.002) or sham cTBS ($t_{11} = 2.99$, p=0.009). There was no significant difference in phonological response times for aIFG and sham cTBS (p=0.95). In contrast, semantic response speed was significantly prolonged with cTBS over aIFG relative to cTBS over pIFG ($t_{11} = 2.89$, p=0.01) or sham cTBS ($t_{11} = 2.23$, p=0.048, does not survive a Bonferroni-Holm correction). These results confirm the functional relevance of left aIFG and pIFG for semantic and phonological processing, respectively. Additionally, a main effect of task ($F_{1,11} = 30.77$, p=0.0001) revealed overall longer response times for phonological than semantic decisions independent of the

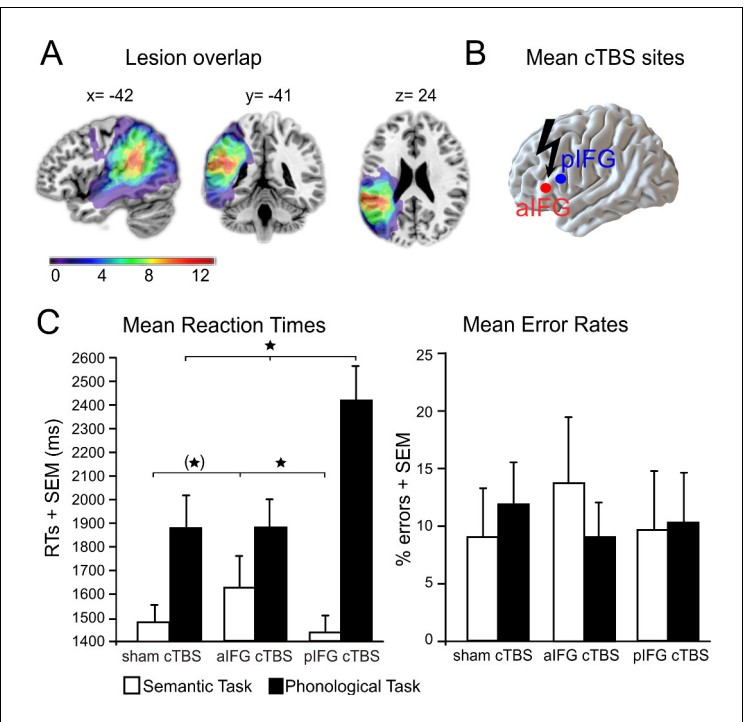

**Figure 2.** Behavioral results. (**A**) Lesion overlap for all patients. All patients had lesions in their left temporo-parietal cortex with the strongest overlap in the supramarginal gyrus. Note that the frontal cortex was intact in all patients. (**B**) Continuous theta burst stimulation (cTBS) sites over the left anterior and posterior inferior frontal gyrus (a/pIFG). Mean stimulation sites were taken from a previous study and transferred to the individual patient's brain. (**C**) Effects of cTBS on task processing. Left panel: Relative to cTBS over aIFG or sham cTBS, cTBS of pIFG significantly delayed phonological response speed. The opposite pattern was found for semantic processing. Relative to cTBS over pIFG or sham cTBS, cTBS over aIFG significantly delayed semantic response speed. Right panel: Effects of cTBS on error rates were not significant. *p<0.05. (*) does not survive a Bonferroni-Holm correction for multiple comparisons. RTs = reaction times, SEM = standard error of the mean. Coordinates are given in MNI space.

**Table 2.** Behavioral results.

| Task/condition | RTs ± SEM (in ms) | ERs ± SEM (in %) |
|---|---|---|
| *Phonological judgements* | | |
| sham cTBS | 1895 ± 132.54 | 11.28 ± 3.42 |
| cTBS of aIFG | 1888 ± 120.90 | 9.55 ± 2.69 |
| cTBS of pIFG | 2411 ± 151.31 | 9.10 ± 1.97 |
| *Semantic judgements* | | |
| sham cTBS | 1486 ± 112.98 | 9.04 ± 4.69 |
| cTBS of aIFG | 1632 ± 129.79 | 12.95 ± 5.88 |
| cTBS of pIFG | 1452 ± 84.18 | 9.36 ± 4.82 |

RTs = reaction times; ERs = error rates; SEM = standard error of the mean.

cTBS condition. Finally, a main effect of cTBS ($F_{2,22}$ = 4.20, p=0.028) showed overall differences between the cTBS conditions with the strongest impact of cTBS over pIFG on task performance (post-hoc t-test only significant for pIFG >sham cTBS: $t_{11}$ = 2.08; p=0.048, does not survive a Bonferroni-Holm correction; pIFG >aIFG cTBS: p=0.13; aIFG >sham cTBS: p=0.33). This effect was likely driven by the strong effect of cTBS over pIFG on phonological response times, as evident in the significant interaction. We did not find any significant differences in error rates (all p>0.05).

## Functional MRI data

### Task processing in the residual language network

We first investigated task-specific effects on neural activity in the residual network at baseline (i.e., after sham cTBS). During both tasks, patients showed increased activity in a large network of bilateral brain regions, mainly in the frontal cortex, that were previously associated with task effort and language comprehension (*Price, 2010*). These regions included the bilateral supplementary motor areas, bilateral anterior insula and inferior frontal gyrus, the right frontal operculum as well as the right pre- and postcentral gyrus, extending into the parietal cortex (*Figure 3*; *Table 3* for details).

## cTBS decreases task-specific neural activity

Our main interest was to investigate the effects of a virtual lesion over the anterior or posterior IFG on task-related activity during phonological and semantic processing. Indeed, the functional-anatomical double dissociation at the behavioral level was also underpinned by different cTBS effects at the neural level (see *Table 3*). Specifically, during phonological decisions, the comparison of cTBS over pIFG relative to sham cTBS showed a strong decrease of task-related activity in the targeted region (left pIFG: x, y, z = −54, 12, 17; T = 4.97; *Figure 4A*), as well as in the supplementary motor area (x, y, z = 1, 3, 54; T = 4.90) and in the right putamen (x, y, z = 18, 6,–7; T = 4.80).

Likewise, the comparison of cTBS over pIFG relative to cTBS of aIFG also revealed decrease of task-related activity in the stimulated area (left pIFG: x, y, z = −54, 23, 18; T = 4.92).

In contrast, semantic decisions were selectively affected by cTBS over left aIFG (*Table 3*). Compared with sham cTBS, cTBS over aIFG resulted in a strong decrease in task-related activity in a larger network of frontal brain areas, including the left aIFG (x, y, z = −36, 31,–1; T = 4.80; *Figure 4B*), left middle frontal gyrus (x, y, z = −45, 35, 26; T = 4.78) and a number of right-hemispheric frontal regions, including the middle frontal gyrus (x, y, z = 45, 35, 32; T = 5.20), superior frontal gyrus (x, y, z = 9, 35, 41; T = 4.99) and the homologous aIFG region (right aIFG: x, y, z = 40, 37, 3; T = 4.90). Similar results were found for the comparison of cTBS over aIFG relative to cTBS over pIFG. Again, cTBS over aIFG resulted in decreased activity in left aIFG (x, y, z = −38, 31, 1; T = 4.78), right middle frontal gyrus (x, y, z = 45, 30, 28; T = 5.0) and right aIFG (x, y, z = 42, 38, 4; T = 4.80). We did not find any brain regions that showed significantly *increased* task-related activity after cTBS over either the aIFG or pIFG during semantic or phonological processing.

## Delayed phonological response speed is associated with the upregulation of the right supramarginal gyrus

We used regression analyses in SPM to test whether the task-specific delay in phonological or semantic response speed would be correlated with changes in task-related activity. To this end, we calculated the difference in response speed for cTBS over either pIFG and sham cTBS (phonological response speed) or aIFG and sham cTBS (semantic response speed). We found that the delay in phonological response speed was positively correlated with the individual upregulation of the right SMG and adjacent postcentral gyrus (x, y, z = 54,–19, 29; T = 6.53) after cTBS over pIFG relative to sham cTBS (*Figure 5A*). Note that this effect remained significant after adjusting for the lesion volume. This shows that those patients who had the strongest delay in phonological response speed after cTBS over pIFG showed the strongest upregulation of the contralesional right SMG, likely in an attempt to compensate for the cTBS-induced disruption of the phonological system. We did not find any significant correlation for semantic decisions.

## Integrity of the right superior longitudinal fasciculus is negatively correlated with phonological response delay

To further explore the potential compensatory capacity of the right parietal cortex after disruption of the left-hemispheric phonological network, we explored whether the individual tract integrity in the underlying white matter tract that connects the right parietal lobe and the inferior frontal cortex, i.e., the right SLF (*Kaplan et al., 2010*), would be correlated with the individual virtual lesion effect. To this end, we correlated the mean fractional anisotropy for each patient within a white matter mask of the right SLF with the individual phonological cTBS-induced response delay after pIFG relative to sham cTBS (*Figure 5B*, upper panel). Indeed, we found a significant negative relationship, indicating that patients who had a stronger tract integrity of the right SLF were less affected by the virtual lesion and might thus have shown a better compensation by recruiting processing resources in the right SMG via the intact right SLF ($R^2$ = 0.42; p=0.02; two-tailed; *Figure 5B*). This effect was anatomically and functionally specific as there was no correlation between phonological response delay and fractional anisotropy in the neighboring IFOF, or between tract integrity and semantic response delay (all p>0.05).

## Discussion

This study represents the first investigation of short-term reorganization in the lesioned language network. Applying focal virtual lesions over the left anterior and posterior inferior frontal gyrus in patients with chronic stroke-induced lesions in the left temporo-parietal cortex, we show task-specific perturbation effects in different networks for language comprehension. Importantly, patients were selected based on their lesion patterns, including the left temporo-parietal cortex but sparing left frontal cortex. This enabled us to investigate if an additional virtual lesion induced by inhibitory continuous theta burst stimulation disrupts language processing and results in an upregulation of contralesional right hemisphere areas, thereby elucidating effects of short-term plasticity. Our results provide evidence for a functional-anatomical double dissociation in the left IFG. The first main finding was that perturbation of a key region for phonological processing in the left pIFG resulted in a strong task-specific delay of phonological response speed. This was accompanied by a decrease in task-related activity in the stimulated area. More notably, the individual cTBS-induced response delay was positively correlated with the upregulation of the contralesional right SMG, likely reflecting an attempt to compensate for the cTBS-induced disruption. These effects were task-specific, as cTBS over left pIFG did not affect semantic processing. Moreover, anatomical specificity was confirmed since cTBS over left aIFG did not affect phonological processing. Finally, individual tract integrity of the right SLF was negatively correlated with the delay in response speed. This indicates that patients with stronger tract integrity of the right SLF show higher resilience against the virtual lesion. Together, our results suggest a compensatory recruitment of right-hemispheric regions in the lesioned language network in patients with temporo-parietal lesions.

In contrast to the observed patterns in the phonological task, semantic processing was selectively affected after cTBS over left aIFG. We found a task-specific delay of semantic decisions, which was paralleled by a large-scale inhibitory network effect after cTBS over aIFG. However, we did not

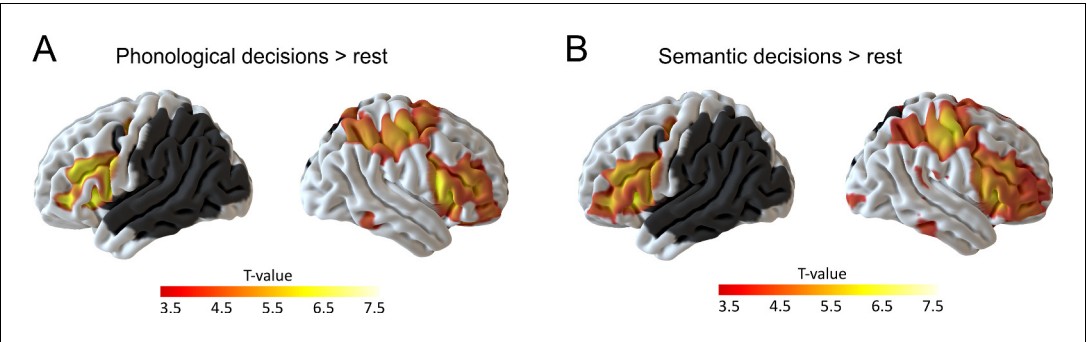

**Figure 3.** Residual language network after sham continuous theta burst stimulation. (**A**) Phonological decisions compared to rest and (**B**) semantic decisions compared to rest. Voxels with at least one lesion are masked in dark grey and were excluded from the analysis. Results are shown at p<0.001 uncorrected for display reasons.

observe any (compensatory) upregulation of homologous right-hemispheric regions for the semantic task.

Together, our findings point towards a strong anatomical and functional specificity of the observed perturbation effects and the resulting short-term reorganization and provide important new insights into the compensatory potential of the lesioned posterior (temporo-parietal) language network. We wish to emphasize that virtual lesions are transient and focal, and particularly suited to investigate immediate short-term plasticity (*Rossi and Rossini, 2004*; *Siebner and Rothwell, 2003*). We reasoned that the lesioned language network should show increased sensitivity to a virtual lesion of a task-specific key region. Our hypothesis was that after challenging the lesioned language network by perturbation of a frontal key language region, we would observe a compensatory shift of task-related activity to the right hemisphere. Since participants only showed mild aphasic symptoms at the time of the study, we infer that our virtual lesion unmasked deficits of the stroke-induced chronic lesion, largely compensated by adaptive plasticity across the course of recovery. Note that on average, we did not find significantly increased activity in the right SMG for the direct comparison of pIFG relative to sham cTBS. Yet, the individual cTBS-induced increase in phonological response speed correlated with the upregulation of the contralesional homologue. This discrepancy likely reflects a threshold phenomenon. When lowering the threshold, task-related activity in right SMG was increased after cTBS of left pIFG relative to sham stimulation, which was mainly driven by those individuals who showed the strongest increase in response speed after cTBS. Together, these results most likely reflect an attempt to compensate for the large-scale disruption of the left hemisphere induced by the 'double lesion', which would support the hypothesis of a compensatory role of the lesion homologue. Specifically, we argue that the acute cTBS-induced increase in the overall lesion load increased processing demands, thereby delaying phonological response speed and necessitating the recruitment of the right hemisphere lesion homologue. This upregulation may have assisted to maintain task processing as indicated by the overall relatively high level of task accuracy. Consequently, we believe that the observed correlation between cTBS-induced response delay and upregulation of the lesion homologue most likely reflects partial compensation by recruitment of a right hemisphere area. While the role of the right hemisphere in language recovery after stroke remains highly controversial, our results speak for an adaptive contribution, at least with respect to short-term adaptation for phonological processing in patients with left temporo-parietal lesions. Indeed, a beneficial role of right-hemispheric regions in language recovery after stroke was suggested in a number of previous studies (*Hamilton et al., 2011*; *Saur et al., 2006*; *Turkeltaub, 2015*). Accordingly, increased activity in a large network of bilateral frontal regions was already observed after sham cTBS in our study. Note that the observed task-specific recruitment of brain regions after sham cTBS was largely similar to that observed in our previous study on healthy volunteers that used the same tasks and comparisons (*Hartwigsen et al., 2017*), except for the absence of any activity in left temporo-parietal regions in the present study that can be explained by the lesion site in our sample of patients.

**Table 3.** Task-related changes in neural activity.

| Region | Side | MNI coordinates (x, y, z) | | | T | Cluster size |
|---|---|---|---|---|---|---|
| *Phonological decisions > rest after sham cTBS* | | | | | | |
| Supplementary motor area | L/R | −3 | 2 | 56 | 8.57 | 619 |
| Supplementary motor area | L | −6 | 11 | 47 | 8.15 | subcluster |
| Supplementary motor area | R | 9 | 17 | 44 | 6.90 | subcluster |
| anterior insula | L | −30 | 26 | 2 | 8.28 | 1527 |
| posterior inferior frontal gyrus (pars opercularis) | L | −49 | 9 | 11 | 8.15 | subcluster |
| posterior inferior frontal gyrus (pars opercularis) | L | −54 | 14 | 1 | 6.97 | subcluster |
| Frontal operculum | R | 42 | 20 | 5 | 6.61 | 532 |
| Posterior inferior frontal gyrus (pars opercularis) | R | 48 | 11 | 17 | 6.47 | subcluster |
| anterior insula | R | 33 | 23 | 5 | 6.32 | subcluster |
| Precentral gyrus (extending to the postcentral gyrus and parietal cortex) | R | 33 | −22 | 53 | 5.90 | 95 |
| *Semantic decisions > rest* | | | | | | |
| Supplementary motor area | L/R | −6 | 14 | 47 | 7.54 | 708 |
| Supplementary motor area | L | −3 | −1 | 56 | 6.87 | subcluster |
| Supplementary motor area | R | 9 | 17 | 41 | 6.69 | subcluster |
| anterior insula | L | −30 | 26 | 2 | 6.79 | 1054 |
| anterior inferior frontal gyrus (pars orbitalis / triangularis) | L | −46 | 42 | −5 | 6.01 | subcluster |
| posterior inferior frontal gyrus (pars opercularis) | L | −50 | 10 | 10 | 5.91 | subcluster |
| Precentral gyrus | R | 39 | −19 | 56 | 6.62 | 318 |
| Postcentral gyrus (extending to the parietal cortex) | R | 48 | −22 | 53 | 4.91 | subcluster |
| Frontal operculum | R | 39 | 23 | 2 | 6.05 | 299 |
| Cerebellum | L | −27 | −52 | −25 | 5.98 | 98 |
| *Phonological decisions: sham cTBS > cTBS of pIFG* | | | | | | |
| posterior inferior frontal gyurs (pars opercularis) | L | −54 | 12 | 17 | 4.97 | 64 |
| Supplementary motor area | R | 1 | 3 | 54 | 4.90 | 60 |
| Putamen | R | 18 | 8 | −7 | 4.80 | 35 |
| *Phonological decisions: cTBS of aIFG > cTBS of pIFG* | | | | | | |
| posterior inferior frontal gyrus (pars opercularis) | L | −54 | 23 | 18 | 4.92 | 60 |
| *Semantic decisions: sham cTBS > cTBS of aIFG* | | | | | | |
| Middle frontal gyrus | R | 45 | 35 | 32 | 5.20 | 347 |
| Inferior frontal gyrus (pars orbitalis) / insula | R | 40 | 37 | 3 | 4.90 | subcluster |
| Superior frontal gyrus | R | 9 | 35 | 41 | 4.99 | 158 |
| Inferior frontal gyrus (pars orbitalis) / insula | L | −36 | 31 | −1 | 4.80 | 65 |
| Middle frontal gyrus | L | −45 | 35 | 26 | 4.78 | 60 |
| *Semantic decisions: cTBS of pIFG > cTBS of aIFG* | | | | | | |
| Middle frontal gyrus | R | 45 | 30 | 28 | 5.00 | 242 |
| Inferior frontal gyrus (pars orbitalis) / insula | R | 42 | 38 | 4 | 4.80 | subcluster |
| Inferior frontal gyrus (pars orbitalis) / insula | L | −38 | 31 | 1 | 4.78 | 60 |

$p<0.05$, FWE corrected at the cluster level.

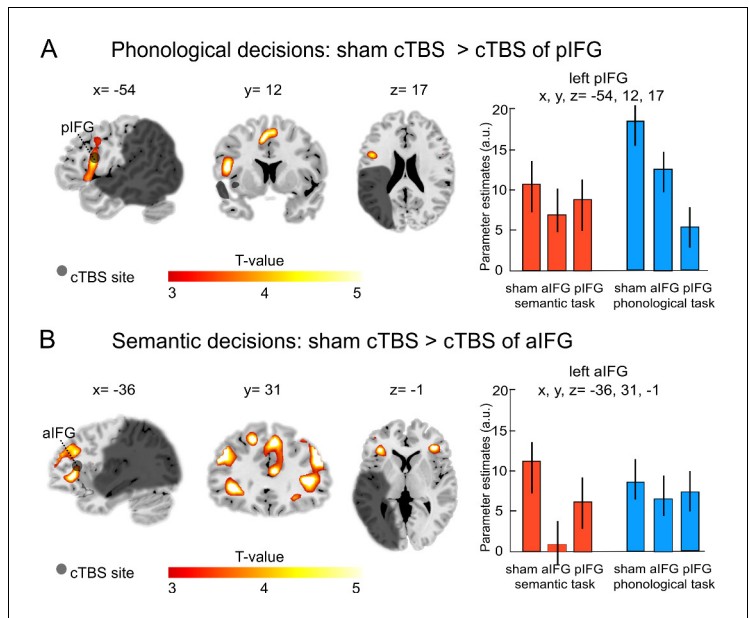

**Figure 4.** Task-specific continuous theta burst stimulation (cTBS) effects on word decisions. (**A**) Phonological decisions. Relative to sham cTBS, cTBS of pIFG significantly decreased neural activity at the stimulated area. (**B**) Semantic decisions. Relative to sham cTBS, cTBS of aIFG significantly decreased neural activity in a larger network, including the bilateral anterior insula and adjacent aIFG. Right panels display the respective parameter estimates (arbitrary units) for the different cTBS conditions that were extracted at the respective mean group peak coordinates from the effect of interest for each task condition against rest. p<0.001 uncorrected for display reasons. a/pIFG = anterior/posterior inferior frontal gyrus. Coordinates are given in MNI space.

However, the specific contribution of the lesion homologue was selectively found after perturbation of an intact node in the phonological network, indicating a high degree of specificity of this effect. A likely explanation for these findings is that the virtual lesion of the pIFG unmasked the contribution of the right SMG to phonological decisions (see *Sack et al., 2005* for a similar ratio). These results are noteworthy because they show that the lesioned network preserves its flexibility to partially compensate for a focal perturbation of a key region in the left-hemispheric language network. Interestingly, we did not observe compensatory increases in right temporo-parietal activity after left temporo-parietal lesions in our recent longitudinal study on language recovery after stroke (*Stockert et al., 2020*). It is likely that the additional TMS-induced perturbation of the left frontal cortex in the present study was necessary to unmask the role of the right hemisphere.

Based on the controversy over the beneficial vs. maladaptive role of the right hemisphere in language recovery after stroke in the literature (e.g. *Crinion and Price, 2005*; *Heiss et al., 2013*; *Naeser et al., 2005*; *Naeser et al., 2011*; *Robson et al., 2014*), an alternative hypothesis is that the upregulation of the right SMG might reflect a cTBS-induced disinhibition of the right hemisphere that is deleterious and causally related to weaker performance. However, we think that this alternative hypothesis is unlikely to explain our findings for the following reasons: First, if our findings mainly reflected a disinhibition, resulting from a cTBS-induced reduction in the transcallosal inhibition from the left to the right hemisphere, then one would have expected a strong upregulation of the stimulation homologue in the right posterior IFG. This is not congruent with our findings. Secondly, a maladaptive role of the right hemisphere would not be congruent with the observed positive correlation between the individual tract integrity in the right SLF and the individual performance in the phonological task, linking stronger tract integrity with a stronger resilience of the phonological network. Finally, a previous study in healthy volunteers showed that TMS-induced perturbation of the right SMG decreased response efficiency in the same syllable judgement task that was used in the present study, providing evidence for a functional role of the right SMG in phonological (working memory) processes (*Hartwigsen et al., 2010*). Together, our results speak more in favor of a compensatory role of the right lesion homologue in the parietal cortex. Yet, to prove the functional

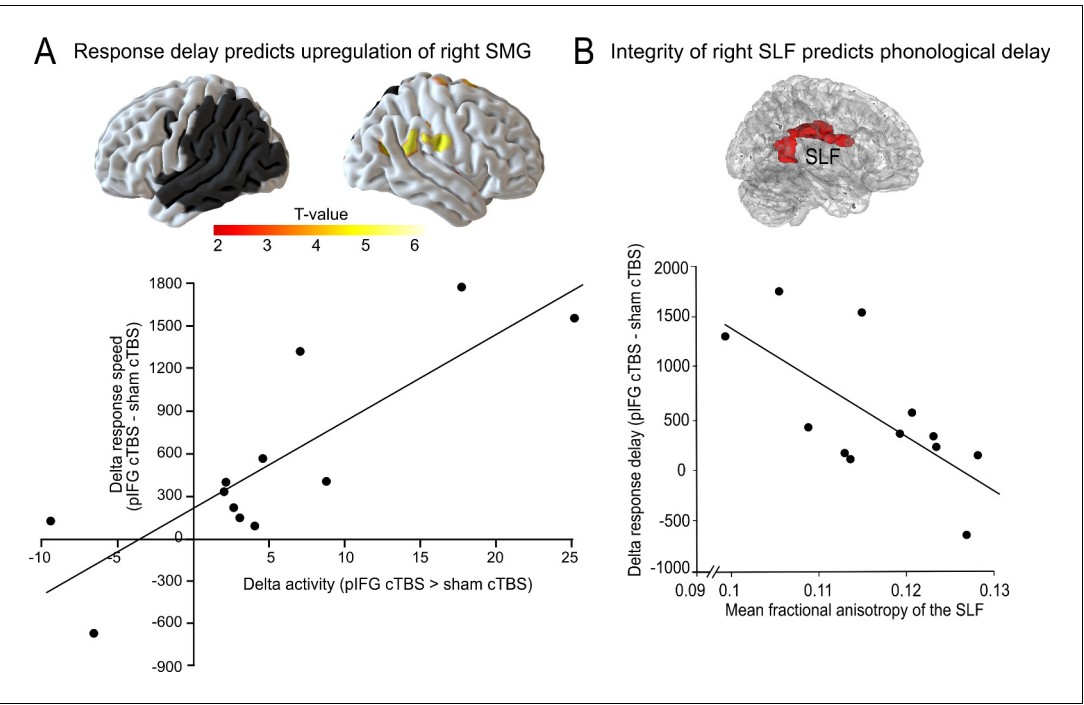

**Figure 5.** Compensatory effects during phonological processing. (**A**) Regression analysis. The individual delay in phonological response speed after effective continuous theta burst stimulation (cTBS) over pIFG vs. sham cTBS (pIFG – sham cTBS) was correlated with the upregulation of the contralesional right supramarginal gyrus (SMG; shown at p<0.001 uncorrected for display reasons). (**B**) Correlation between the individual mean fractional anisotropy (FA) in the right superior longitudinal fasciculus (SLF) and the behavioural cTBS effect. Upper panel: 3D rendering generated with FSLview showing the region of interest in the right SLF that was used to extract the mean FA. The SLF ROI was obtained from the Juelich Histological atlas. Lower panel: Regression analysis. The individual FA for the right SLF was negatively correlated with the relative increase in the individual mean reaction times for the phonological task after cTBS of pIFG relative to sham cTBS.

relevance of the lesion homologue for phonological processing, an additional experiment would be required that should target both left pIFG and right SMG in the same patients with inhibitory TMS. If the notion of a beneficial, compensatory reorganization in the right hemisphere holds true, this should result in stronger performance impairment (i.e., decreased phonological accuracy). In contrast, if the observed upregulation reflected disinhibition, perturbation of the right SMG should result in behavioral improvements. However, such a double perturbation experiment might be difficult to realize in patients with brain lesions due to an increased risk for side effects when targeting two nodes in the lesioned brain with inhibitory TMS.

The identified structural predictor of differences in resilience to virtual lesions suggests that differences in the individual structural network architecture modulate the compensatory potential of the non-dominant hemisphere, enabling large-scale interactions in right-hemispheric homologous language regions. This notion is supported by a previous study in patients with aphasia after left-hemispheric stroke that reported a correlation between increased grey or white matter volume in the contralesional right SMG and preserved speech abilities (*Balaev et al., 2016*). In our study, premorbid individual variance rather than a compensatory change in white matter integrity after the stroke-induced lesion appears to be the most likely factor that affected the individual ability for short-term compensation. Yet, to disentangle these two explanations, a longitudinal study that addresses potential changes in tract integrity across the time period of language recovery would be needed. In any case, the previous and present findings speak in favour of a compensatory role of the lesion homologue in the right inferior parietal cortex.

While our study is experimental in nature, these results are relevant to clinical perspectives such as the reorganization after a second stroke and the development of neurophysiologically-based therapeutic approaches. Our results suggest that with increased task demands (i.e., with large lesions),

the facilitation of lesion homologues with non-invasive brain stimulation may be an avenue for thera-peutic intervention. Indeed, right-hemispheric recruitment might become increasingly relevant if the left-hemispheric network lacks the capacity to recover due to large lesions (*Heiss and Thiel, 2006*). While an increasing number of studies use non-invasive brain stimulation to support language ther-apy (e.g. *Coslett, 2016*; *Otal et al., 2015*), most of these studies focus on facilitation of perilesional regions or inhibition of contralesional frontal regions. To date, only a few studies applied facilitatory stimulation over right-hemispheric regions (*Flöel et al., 2011*; *Vines et al., 2011*). Based on our find-ings, we encourage such applications especially for challenging conditions. However, we wish to emphasize that this might not generalize to all lesion patterns and aphasia types since some studies also reported language improvement after *inhibition* of right hemispheric homologues (see *Naeser et al., 2005*; *Naeser et al., 2011*). Indeed, the beneficial or maladaptive role of the right hemisphere in language recovery may strongly depend on the lesion site (e.g. frontal vs. temporo-parietal stroke) and size and the aphasia type (e.g. *Heiss and Thiel, 2006*; *Naeser et al., 2011*). The observed short-term reorganization in the right-hemisphere in our study might be best comparable with an early beneficial recruitment of right-hemispheric language homologues in post-stroke apha-sia (*Saur et al., 2006*).

In contrast to the focal virtual lesion effect on phonological decisions, we observed widespread inhibition of bilateral frontal regions during semantic decisions after cTBS over left aIFG. The observed inhibition included bilateral domain-general and executive semantic regions, some of which were previously associated with cognitive control and executive processing (*Baumgaertner et al., 2013*; *Geranmayeh et al., 2017*). The fact that a focal perturbation of the left IFG was sufficient to result in widespread bilateral activity decreases points towards strong remote effects of plasticity-inducing TMS protocols as demonstrated in previous studies in the semantic net-work (*Binney and Ralph, 2015*; *Hallam et al., 2016*; *Hartwigsen et al., 2017*). The behavioural dis-ruption in combination with a bilateral inhibition of task-related activity in our study is further congruent with previous lesion studies indicating that semantic impairment is often associated with bilateral disruption of semantic key regions (*Bi et al., 2011*; *Lambon Ralph et al., 2012*). Together, these results speak in favour of a bilateral organization of the (executive) semantic system with densely connected key areas (*Jung and Lambon Ralph, 2016*; *Thompson et al., 2016*). The strong inhibitory network effect is further in line with our previous study in healthy volunteers that used a similar task (*Hartwigsen et al., 2017*). However, the absence of compensatory upregulation of other (semantic) network nodes is surprising given that previous rTMS studies in healthy volunteers dem-onstrated increased task-related activity in homologous right-hemispheric regions (*Andoh and Paus, 2011*; *Jung and Lambon Ralph, 2016*) or neighbouring left-hemispheric regions (*Hallam et al., 2016*; *Hartwigsen et al., 2017*) after perturbation of semantic key regions. The absence of short-term reorganization in our study is most likely explained by the above-discussed widespread inhibi-tion effect in both domain-specific semantic and domain-general areas in the frontal cortex. Note that we did not observe any (compensatory) upregulation of the intact ventrolateral ATL either, a region that was previously identified as key area for semantic processing (e.g. *Jung and Lambon Ralph, 2016*). The absence of its recruitment in our study might have resulted from the use of a 'standard' fMRI sequence that lacked sensitivity to capture signal changes in the anterior temporal lobe (see *Halai et al., 2014*). In summary, the behavioural perturbation effect on semantic decisions in our patients speaks for a global inhibitory network effect, supporting the view that the lesioned language network is more susceptible to virtual lesion effects.

The discrepancy in short-term plasticity between the above-mentioned virtual lesion studies in healthy volunteers and the present study is likely explained by a number of differences between studies. Most importantly, our findings may indicate that adaptive plasticity after a virtual lesion is qualitatively different in the reorganized language network. Another explanation for the observed discrepancies relates to the site of the virtual lesion. Here, we targeted key language regions in the inferior frontal gyrus. Across both tasks, we found strong virtual lesion effects at the behavioural and neural level, as evident from the strong inhibition in the targeted area. In contrast, our previous study that used the same tasks in healthy volunteers applied cTBS to different regions in the parietal cortex (*Hartwigsen et al., 2017*). In that study, behavioural perturbation was selectively observed for the phonological but not for the semantic task. Behavioral rTMS effects on semantic processing were only found after *combined* perturbation of left AG and aIFG in our previous work (*Hartwigsen et al., 2016*). Hence, it appears likely that the strong effects found in the reorganized

language network can be explained by the increased lesion load after the 'double lesion' in the left hemisphere. Together, the previous and present results indicate that the network challenge induced by a combination of (virtual) lesions has a strong impact on neural activity and behavioural performance. Despite this increase in the lesion load, task accuracy was not significantly impaired by cTBS in either of the two tasks, suggesting some degree of robustness of the reorganized network against the cTBS effect. This converges with previous studies in healthy participants, which report behavioural TMS effects on response speed rather than accuracy, suggesting that the TMS-induced 'neural noise' is unlikely to completely 'silence' the targeted area (*Miniussi et al., 2013*).

It should be emphasized that the observed functional-anatomic double dissociation in our tasks is congruent with previous evidence from healthy participants that associated left aIFG with semantic and left pIFG with phonological processing (*Gough et al., 2005*; *Romero et al., 2006*; *Vigneau et al., 2006*). Extending previous findings, we show that the task-specific behavioural perturbation effect was underpinned by task-specific inhibition at the respective targeted region that likely represents the neural correlate of the virtual lesion effect.

The present study is limited by the small sample size. However, we wish to emphasize that all patients underwent two active cTBS conditions and a sham condition, allowing for a complete within-subject design and thus direct comparisons between the different cTBS conditions in the same patients. While the present study focused on relatively well-recovered patients, future work should explore whether the degree of language impairment after left temporo-parietal lesions may predict the upregulation of the right SMG. If the notion of a beneficial role of the right SMG holds true, then we would predict that more severe language impairment may require a stronger contribution of the right hemisphere. Notably, two of our patients showed the opposite pattern compared to the average of the group (i.e., decreased activity in the right SMG after cTBS over left pIFG relative to sham stimulation) and one of them had faster phonological response speed after cTBS. Given the small number of observations (2 out of 12), these findings should be taken with caution. Nevertheless, it is interesting to note that both patients were fully recovered at the time of study and had the overall smallest lesion sizes. While lesion size was included in our fMRI design as a covariate, these findings might suggest that the recruitment of the right hemisphere was restricted to patients with larger lesion sizes and relatively stronger language deficits.

In summary, we demonstrate task-specific functional reorganization in the lesioned language network. The upregulation of the contralesional right homologue after a 'double lesion' in the language network implicates that increased task demands induced by neuronal network challenges likely require the contribution of right-hemispheric regions.

## Materials and methods

### Patients

12 right-handed native German-speaking patients with a single chronic embolic stroke (mean age: 58.8 years, range: 43–72 years) and no history of other neurological disorders or TMS contraindications completed our study. The data of two additional patients were excluded since one patient did not come back after the first session and another one did not show any sound-related neural activation in the auditory cortex after sham cTBS. Patients were selected based on their lesion site, including the left (temporo-)parietal cortex while leaving the frontal cortex intact (*Figure 2A* and *Table 1*). All patients were in the chronic phase after stroke (>6 months, mean 37.9, SD = 34.8 months) and initially presented with aphasia but showed (almost) complete language recovery at the time of the experiment. Specifically, 7 patients were diagnosed with residual aphasia, and the remaining 5 patients had no residual aphasia at time of participation (range, 6 to 122 months post-stroke, see *Table 1*).

Written informed consent was obtained before the experiment that was in accordance with the Declaration of Helsinki and approved by the local ethics committee (Medical Faculty at the University of Leipzig).

### Experimental design

This study used a two (*task: semantic vs. phonological decisions*) by three (*cTBS: effective cTBS of aIFG or pIFG and sham cTBS*) factorial within-subject design. In three sessions (inter-session

interval ≥7 days), we applied effective or sham cTBS over left aIFG or pIFG prior to fMRI. As the TMS laboratory was situated close to the MR unit, the average time between stimulation offset and fMRI onset was 6 min. Note that patients were seated in an MR-compatible wheelchair prior to cTBS application to avoid movements after the cTBS intervention and allow for a rapid transfer to the MR scanner.

Each fMRI session was divided into two runs with a short break. During fMRI, patients performed a semantic ('natural or man-made item?') and a phonological word decision task ('2 or 3 syllables?'). Auditory word stimuli were presented in miniblocks of 6 items, separated by resting periods of 16 s. Tasks were held constant within each miniblock and pseudorandomized across blocks. We used identical stimuli for both tasks that were pseudorandomized across conditions and counterbalanced across subjects. Each session consisted of 10 miniblocks per task, leading to a total presentation of 120 stimuli per session. The same stimuli were used in all sessions. Patients were instructed to respond as quickly and accurately as possible by button press with their left middle or index finger. Stimuli were taken from our previous study in healthy participants (*Hartwigsen et al., 2016*). Only highly frequent, unambiguous nouns from the CELEX lexical database for German (Centre for Lexical Information, Max Planck Institute for Psycholinguistics, The Netherlands) were included. In total, 60 two-syllable nouns and 60 three-syllable nouns were selected. All words represented natural or man-made items (i.e., 50% of the 2-syllable words and 50% of the 3-syllable words, respectively).

## Magnetic resonance imaging

MRI data were obtained at a 3-Tesla MRI (Siemens Verio; Siemens, Erlangen). Functional imaging was performed using a gradient EPI sequence ([TR] / [TE]=2520/30 ms, flip angle = 90°, matrix = 64×64 pixel, voxel size = 3×3 x 3 mm; field of view = 192 mm$^2$) with BOLD contrast for the acquisition of T2*-weighted images. 370 volumes consisting of 38 slices were acquired continuously during each run. Additionally, all patients underwent structural MR imaging, including a high-resolution T1-weighted scan (MPRAGE; 170 slices, voxel size = 1×1 x 1.5 mm, matrix = 240×240 pixel, TR = 1.3 s, TE = 3.46 ms) and diffusion tensor imaging (DTI). For DTI analyses, we acquired axial whole brain diffusion weighted images with a double spin echo sequence (60 directions; *b*-value = 1000 s/mm$^2$; 88 slices; voxel size, 1.7 × 1.7 × 1.7 mm, no slice gap; TR = 12.9 s; TE = 100 ms; field of view = 220 × 220 mm$^2$) plus seven volumes with no diffusion weighting (*b-value* = 0 s/mm$^2$) at the beginning of the sequence and interleaved after each block of 10 images.

## Continuous theta burst stimulation

We used neuronavigated cTBS (Brainsight; Rogue Research) based on coregistered individual T1-weighted images to navigate the TMS coil and maintain its exact location throughout sessions. Session order was counterbalanced across patients. cTBS was performed using the mean MNI coordinates for left aIFG (x, y, z = −45, 27, 12 mm) and pIFG (x, y, z = −47, 6, 21 mm) from our previous rTMS study in healthy volunteers (*Hartwigsen et al., 2016*). Individual stimulation sites were determined by calculating the inverse of the normalization transformation (obtained after segmentation and normalization of the individual T1-weighted image) and transforming the coordinates from standard to individual space for each patient.

The TMS coil was positioned tangentially on the head with the handle pointing at 45° to the sagittal plane. In each session, we applied cTBS (600 stimuli at 50 Hz in trains of three stimuli at an inter-burst interval of 200 ms for 40 s) over either left aIFG or pIFG. We used a stimulation intensity of 80% of the individual active motor threshold (AMT). AMT was defined as the lowest stimulus intensity producing a motor evoked potential of 150–200 μV in the tonically active first dorsal interosseus muscle (20% of maximum contraction). A figure-of-eight-shaped coil (type CB-60) connected to a MagPro X100 stimulator (MagVenture, Farum, Denmark) was used for effective cTBS. For sham cTBS, we used a figure-of-eight shaped placebo coil that creates an identical sound level but provides an effective field reduction of 80%. Half of the patients received sham cTBS over aIFG and pIFG, respectively. The overall application of TMS pulses per sessions was well within safety limits and the whole procedure was in accordance with the current safety guidelines (*Rossi et al., 2009*).

## Data analyses
### Behavioral data
After a short training that did not contain any items used in the main experiments, all patients were able to perform both tasks. Reaction times for correct responses were analyzed with a repeated-measures ANOVA. Error rates were separately investigated with non-parametric Wilcoxon signed-rank tests. An $\alpha$-level < 0.05 (two-tailed) was considered significant for all comparisons. Two-tailed post-hoc paired t-tests with Bonferroni-Holm corrections for multiple comparisons explored differences among conditions. All statistical analyses were performed with SPSS software (version 22, Chicago, IL, USA).

### fMRI data
Task-related changes in the blood oxygenation level-dependent signal were analyzed with SPM 12 (Wellcome Trust Centre for Neuroimaging; www.fil.ion.ucl.ac.uk/spm/) implemented in Matlab (The Mathworks, Inc, Natick, MA). Preprocessing of the fMRI data included slice timing, data realignment, coregistration of the T1-weighted and functional EPI images, segmentation, normalization into standard space and smoothing with an isotropic 8 mm FWHM Gaussian kernel. Statistical analyses were performed in two steps. The individual first level included regressors for each task condition, the instruction and the realignment parameters. All onsets in each regressor were convolved with a canonical hemodynamic response function as implemented in SPM12. Voxel-wise regression coefficients for all conditions were estimated using the least squares method within SPM12, and statistical parametric maps of the t-statistic were generated from each condition (i.e., effects of tasks vs. rest under the different cTBS conditions). The data for the second level comprised pooled parameter estimates for each of these contrasts across participants in a random-effects analysis using a flexible factorial within-subject ANOVA design. We used the Restricted Maximum Likelihood method in the SPM12 design specification for sphericity correction at the second-level inference. Additionally, we performed regression analyses to delineate regions where cTBS-disruption of task performance (i.e., increases in reaction time) predicted an upregulation of activity for each task. Note that all voxels lesioned in at least one patient were excluded from further analyses. All comparisons were thresholded at an initial significance level of p<0.001, uncorrected and corrected for multiple comparisons using a p<0.05 FWE-corrected at the cluster level. The SPM anatomy toolbox and the WFU PickAtlas Tool (Wake Forest University of School of Medicine) were used for anatomical localization.

## Tract-based analysis of white matter integrity
To extract the individual fractional anisotropy in dorsal and ventral fiber tracts in the intact right hemisphere, we used the Leipzig Image Processing and Statistical Inference Algorithms (https://www.cbs.mpg.de/institute/software/lipsia) and the FMRIB's diffusion toolbox (FDT, Oxford Centre for Functional Magnetic Resonance Imaging of the Brain Diffusion Toolbox, Version 5.0) for tract-based spatial statistics (TBSS). Preprocessing of the data included motion correction based on the seven b0 images and global registration to the T1 anatomy. Subsequently, a diffusion tensor model was fitted to the preprocessed data (*Behrens et al., 2007*).

Fractional anisotropy (FA) maps were calculated for each patient. Here, the degree of anisotropy indicates the integrity and density of oriented structures in the tissue of interest. We used anatomical masks for the right superior longitudinal fasciculus (SLF) and inferior fronto-occipital fasciculus (IFOF) based on the Juelich Histological atlas (*Bürgel et al., 2006*) to extract tract-specific FA values in each patient by averaging TBSS-preprocessed, skeletonized FA data within these regions of interest (ROIs). Mean FA values were then correlated with the behavioural data. The choice of these ROIs was based on the assumption that these tracts most likely represent the homologue tracts of the dorsal (SLF) and ventral (IFOF) pathways for phonological and semantic processing as described for the left hemisphere (*Saur et al., 2008*) that should connect our cTBS sites in the left IFG with the angular and supramarginal gyrus, respectively.

## Acknowledgements

This study was supported by the Deutsche Forschungsgemeinschaft (HA 6314/1–1, HA and HA 6314/4–1 to GH and SA 1723/5-1 to DS and JK) and by the Max Planck Society. DS is supported by the James S McDonnell Foundation.

## Additional information

### Funding

| Funder | Grant reference number | Author |
|---|---|---|
| Deutsche Forschungsge-meinschaft | 6314/1-1 | Gesa Hartwigsen |
| Deutsche Forschungsge-meinschaft | 6314/3-1 | Gesa Hartwigsen |
| Deutsche Forschungsge-meinschaft | 6314/4-1 | Gesa Hartwigsen |
| Deutsche Forschungsge-meinschaft | SA 1723/5-1 | Dorothee Saur Julian Klingbeil |
| James S. McDonnell Founda-tion | Understanding Human Cognition ("Function, dysfunction and repair of language networks") | Dorothee Saur |

The funders had no role in study design, data collection and interpretation, or the decision to submit the work for publication.

### Author contributions

Gesa Hartwigsen, Conceptualization, Data curation, Formal analysis, Supervision, Funding acquisition, Investigation, Visualization, Methodology, Writing - original draft, Project administration, Writing - review and editing; Anika Stockert, Data curation, Formal analysis, Visualization, Methodology, Writing - review and editing; Louise Charpentier, Data curation, Methodology; Max Wawrzyniak, Data curation, Formal analysis, Investigation, Methodology, Writing - review and editing; Julian Klingbeil, Katrin Wrede, Investigation; Hellmuth Obrig, Dorothee Saur, Conceptualization, Investigation, Writing - review and editing

### Author ORCIDs

Gesa Hartwigsen (iD) https://orcid.org/0000-0002-8084-1330

### Ethics

Human subjects: Written informed consent was obtained before the experiment that was in accordance with the Declaration of Helsinki and approved by the local ethics committee (Medical Faculty at the University of Leipzig, no. 328-10-08112010).

### Decision letter and Author response

Decision letter https://doi.org/10.7554/eLife.54277.sa1
Author response https://doi.org/10.7554/eLife.54277.sa2

## Additional files

### Supplementary files
• Transparent reporting form

## Data availability

Data cannot be shared completely due to the data protection agreement (approved by the local ethics committee) signed by the participants. Some processed datasets can be found at https://osf.io/4bhrt/.

The following dataset was generated:

| Author(s) | Year | Dataset title | Dataset URL | Database and Identifier |
|---|---|---|---|---|
| Hartwigsen G | 2020 | tbss statistics | https://doi.org/10.17605/OSF.IO/4BHRT | Open Science Framework, 10.17605/OSF.IO/4BHRT |

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
