## [Decision Letter]

Thank you for submitting your article "Short-term reorganization in the lesioned language network" for consideration by *eLife*. Your article has been reviewed by two peer reviewers, including Argye E Hillis as the Reviewing Editor and Reviewer #1, and the evaluation has been overseen by Floris de Lange as the Senior Editor. The following individual involved in review of your submission has agreed to reveal their identity: Dirk Den Ouden (Reviewer #2).

The reviewers have discussed the reviews with one another and the Reviewing Editor has drafted this decision to help you prepare a revised submission.

Title: Unless you are able to provide strong arguments for your position that these results primarily reflect a (beneficial) compensatory role for RH, you might consider a title with broader appeal, such as, "Short-term modulation of the lesioned language network"

Summary:

This paper reports on a study in 12 post-aphasic (partially or fully recovered) patients with left-hemisphere temporoparietal lesions, who performed semantic and phonological judgment tasks after inhibitory brain stimulation via continuous theta burst stimulation (cTBS) to the intact anterior or posterior inferior frontal gyrus (IFG). Results confirm a functional dissociation between anterior and posterior IFG in these patients, with aIFG supporting semantic processing and pIFG supporting phonological processing, as reflected in delayed response times after inhibitory stimulation. Delayed phonological processing was also correlated with upregulation of the BOLD response in the right-hemisphere supramarginal gyrus, as well as with structural integrity of the right-hemisphere superior longitudinal fascicle. The authors conclude that this suggests or confirms a short-term compensatory role for the right-hemisphere in stroke survivors with left-lesioned brains who have recovered from aphasia.

We agree that your submission is a well-written manuscript with novel findings that may impact our understanding of recovery from aphasia, if the interpretation of the results is modified or better justified.

Essential revisions:

1) The primary concern of both reviewers is that your findings do not clearly support your conclusions. The results provide some evidence that individual upregulation of right SMG is associated with increased delay in phonological response. "We found that the delay in phonological response speed was positively correlated with the individual upregulation of the right SMG and adjacent postcentral gyrus (x, y, z = 54, -19, 29; T = 6.53) after cTBSover pIFG relative to sham cTBS(R2=0.71; β=0.87,p<0.001; two-tailed), (Figure 5A)" and "The individual delay in phonological response speed after effective continuous theta burst stimulation(cTBS)over pIFG vs. sham cTBS(pIFG -sham cTBS) was correlated with the upregulation of the contralesional right supramarginal gyrus."

However, we did not understand how these results support the conclusion that the right SMG activity is adaptive (rather than maladaptive). We also do not understand the conclusions, "These findings encourage the application of facilitatory non-invasive brain stimulation to right-hemispheric lesion homologues to support language recovery after left-hemispheric lesions in temporo-parietal regions." If greater activity in right SMG is associated with worse phonological processing (greater delay), would it not make more sense to inhibit right SMG activity? Please clarify if we are misunderstanding the data.

2) Is your assumption that the SLF was strengthened compensatorily as a result of the temporoparietal lesion in these patients, or that the variance in SLF integrity primarily reflects premorbid individual variance, affecting an individual's ability for short-term compensation?

3) It would be worth commenting on the fact 2 patients showed the pattern opposite to the average for the group – decreased activity in right SMG with the virtual lesion, and one of these showed a faster phonological response time (shorter delay).

4) It is also interesting that on average you did not find any brain regions that showed significantly increased task-related activity after cTBS over either the aIFG or pIFG during semantic or phonological processing. So, on average, there was not increased right hemisphere activity; but a few individuals showed significantly increased right SMG activity (which correlated with greater delays in phonological decisions). This finding should be clarified and discussed.

5) In contrast to the focal virtual lesion effect on phonological decisions, they observed widespread inhibition of bilateral frontal regions during semantic decisions after cTBS over left aIFG. This result is consistent with studies that indicate that inhibition or lesions to both frontal/anterior temporal lobes is necessary to produce impairment in semantic processing. This point might be further discussed.

6) Were results different for patients who had fully recovered versus those who had residual deficits? It is a strength that both types of patients are included, but some mention should be made if there were differences (or not).

---

## [Author Response]

Essential revisions:1) The primary concern of both reviewers is that your findings do not clearly support your conclusions. The results provide some evidence that individual upregulation of right SMG is associated with increased delay in phonological response. "We found that the delay in phonological response speed was positively correlated with the individual upregulation of the right SMG and adjacent postcentral gyrus (x, y, z=54, -19, 29; T=6.53) after cTBSover pIFG relative to sham cTBS(R2=0.71; β=0.87, p<0.001; two-tailed), (Figure 5A)" and "The individual delay in phonological response speed after effective continuous theta burst stimulation(cTBS)over pIFG vs. sham cTBS(pIFG -sham cTBS) was correlated with the upregulation of the contralesional right supramarginal gyrus."However, we did not understand how these results support the conclusion that the right SMG activity is adaptive (rather than maladaptive). We also do not understand the conclusions, "These findings encourage the application of facilitatory non-invasive brain stimulation to right-hemispheric lesion homologues to support language recovery after left-hemispheric lesions in temporo-parietal regions." If greater activity in right SMG is associated with worse phonological processing (greater delay), would it not make more sense to inhibit right SMG activity? Please clarify if we are misunderstanding the data.

We apologize for the confusion. Our main line of argumentation is that the observed association between the upregulation of the lesion homologue in the right SMG and the TMS-induced delay in the phonological response speed likely reflects an attempt to compensate for the disruption induced by TMS. Indeed, the functional relevance of intact right SMG function for phonological decisions was demonstrated in a previous study with healthy volunteers (Hartwigsen et al., 2010). However, we agree with the editor and reviewer that this interpretation remains speculative. The ultimate proof that this upregulation reflects an adaptive rather than maladaptive upregulation would require an additional experiment targeting the "reorganized" right SMG with inhibitory TMS. If the combination of left IFG perturbation and right SMG disruption in the lesioned language network would result in a stronger virtual lesion effect (also affecting phonological accuracy), this would show that both regions make a functional contribution to phonological processing in the lesioned brain (for a similar ratio, see Hartwigsen et al., 2016) and that the observed upregulation of the right SMG was indeed adaptive. However, such a double-lesion approach would be risky in patients with stroke-induced brain lesions.

To provide a more balanced discussion of the alternative interpretations of our findings (i.e., adaptive vs. maladaptive plasticity), we have rewritten parts of the Discussion and toned down some of our conclusions. We have also deleted the statement *"These findings encourage the application of facilitatory non-invasive brain stimulation to right-hemispheric lesion homologues to support language recovery after left-hemispheric lesions in temporo-parietal regions."* at the end of the Discussion section because we agree with the reviewers that this sentence might have overemphasized our findings.

The revised passages of the Discussion read as follows:

"This enabled us to investigate if an additional virtual lesion induced by inhibitory continuous theta burst stimulation disrupts language processing and results in an upregulation of contralesional right hemisphere areas, thereby elucidating effects of short-term plasticity. […] More notably, the individual cTBS-induced response delay was positively correlated with the upregulation of the contralesional right SMG, likely reflecting an attempt to compensate for the cTBS-induced disruption."

"Together, our results suggest a compensatory recruitment of right-hemispheric regions in the lesioned language network in patients with temporo-parietal lesions. […] While the role of the right hemisphere in language recovery after stroke remains highly controversial, our results speak for an adaptive contribution, at least with respect to short-term adaptation for phonological processing in patients with left temporo-parietal lesions."

"Interestingly, we did not observe compensatory increases in right temporo-parietal activity after left temporo-parietal lesions in our recent longitudinal study on language recovery after stroke (Stockert et al., 2020). […] However, such a double perturbation experiment might be difficult to realize in patients with brain lesions due to an increased risk for side effects when targeting two nodes in the lesioned brain with inhibitory TMS."

"In summary, we demonstrate task-specific functional reorganization in the lesioned language network. The upregulation of the contralesional right homologue after a "double lesion" in the language network implicates that increased task demands induced by neuronal network challenges likely require the contribution of right-hemispheric regions."

2) Is your assumption that the SLF was strengthened compensatorily as a result of the temporoparietal lesion in these patients, or that the variance in SLF integrity primarily reflects premorbid individual variance, affecting an individual's ability for short-term compensation?

We apologize for not being clearer about this issue in the previous version of our manuscript. We would indeed argue that premorbid individual variance rather than a compensatory change in white matter integrity after the stroke-induced lesion is the most likely factor that affects the individual ability for short-term compensation. We have included the following passage to the Discussion section to clarify this issue:

"In our study, premorbid individual variance rather than a compensatory change in white matter integrity after the stroke-induced lesion appears to be the most likely factor that affected the individual ability for short-term compensation. […] In any case, the previous and present findings speak in favour of a compensatory role of the lesion homologue in the right inferior parietal cortex."

3) It would be worth commenting on the fact 2 patients showed the pattern opposite to the average for the group – decreased activity in right SMG with the virtual lesion, and one of these showed a faster phonological response time (shorter delay).

As suggested, we have now added a short passage on this observation to the Discussion section. Given that only two out of 12 patients showed a different pattern, we feel that this difference should not be overemphasized. The new passage of the revised manuscript reads as follows:

"Notably, two of our patients showed the opposite pattern compared to the average of the group (i.e., decreased activity in the right SMG after cTBS over left pIFG relative to sham stimulation) and one of them had faster phonological response speed after cTBS. Given the small number of observations (2 out of 12), these findings should be taken with caution. […] While lesion size was included in our fMRI design as a covariate, these findings might suggest that the recruitment of the right hemisphere was restricted to patients with larger lesion sizes and relatively stronger language deficits."

4) It is also interesting that on average you did not find any brain regions that showed significantly increased task-related activity after cTBS over either the aIFG or pIFG during semantic or phonological processing. So, on average, there was not increased right hemisphere activity; but a few individuals showed significantly increased right SMG activity (which correlated with greater delays in phonological decisions). This finding should be clarified and discussed.

Please see also reply to comment 1. We have included the following passage to the Discussion to address this issue: "Note that on average, we did not find significantly increased activity in the right SMG for the direct comparison of pIFG relative to sham cTBS. […] Specifically, we argue that the acute cTBS-induced increase in the overall lesion load increased processing demands, thereby delaying phonological response speed and necessitating the recruitment of the right hemisphere lesion homologue."

5) In contrast to the focal virtual lesion effect on phonological decisions, they observed widespread inhibition of bilateral frontal regions during semantic decisions after cTBS over left aIFG. This result is consistent with studies that indicate that inhibition or lesions to both frontal/anterior temporal lobes is necessary to produce impairment in semantic processing. This point might be further discussed.

This is indeed an interesting point. We have included the following passage to the Discussion section to elaborate on this issue:

"In contrast to the focal virtual lesion effect on phonological decisions, we observed widespread inhibition of bilateral frontal regions during semantic decisions after cTBS over left aIFG. […] Together, these results speak in favour of a bilateral organization of the (executive) semantic system with densely connected key areas (cf. Jung and Lambon Ralph, 2016; Thompson, Henshall, and Jefferies, 2016)."

6) Were results different for patients who had fully recovered versus those who had residual deficits? It is a strength that both types of patients are included, but some mention should be made if there were differences (or not).

This is an interesting issue, please see also response to comment #3 above. Indeed, the five patients who showed no aphasia at the time of study had relatively small lesions. As noted above, two of them had decreased activity in right SMG after cTBS over the left pIFG and one of them further showed a facilitatory behavioural effect on phonological response speed after cTBS. This might suggest that the observed correlation between increased right hemispheric activity in the lesion homologue and behavioural disruption is mediated by aphasia severity. Yet, aside from the two diverging cases, the other three patients with complete recovery showed behavioural and neural patterns that were comparable to the rest of the sample. Consequently, these findings should not be overinterpreted.